# Pulsed electroreduction of low-concentration nitrate to ammonia

Yanmei Huang [1,2,7], Caihong He[3,7], Chuanqi Cheng [4], Shuhe Han[1], Meng He[1], Yuting Wang [1,2], Nannan Meng [1], Bin Zhang [1], Qipeng Lu [3,5] ✉ & Yifu Yu [1,2,6] ✉

Electrocatalytic nitrate ($NO_3^-$) reduction to ammonia (NRA) has emerged as an alternative strategy for effluent treatment and ammonia production. Despite significant advancements that have been achieved in this field, the efficient conversion of low-concentration nitrate to ammonia at low overpotential remains a formidable challenge. This challenge stems from the sluggish reaction kinetics caused by the limited distribution of negatively charged $NO_3^-$ in the vicinity of the working electrode and the competing side reactions. Here, a pulsed potential approach is introduced to overcome these issues. A good NRA performance (Faradaic efficiency: 97.6%, yield rate: 2.7 mmol$^{-1}$ h$^{-1}$ mg$_{Ru}^{-1}$, conversion rate: 96.4%) is achieved for low-concentration (≤10 mM) nitrate reduction, obviously exceeding the potentiostatic test (Faradaic efficiency: 65.8%, yield rate: 1.1 mmol$^{-1}$ h$^{-1}$ mg$_{Ru}^{-1}$, conversion rate: 54.1%). The combined results of in situ characterizations and finite element analysis unveil the performance enhancement mechanism that the periodic appearance of anodic potential can significantly optimize the adsorption configuration of the key *NO intermediate and increase the local $NO_3^-$ concentration. Furthermore, our research implies an effective approach for the rational design and precise manipulation of reaction processes, potentially extending its applicability to a broader range of catalytic applications.

Excessive nitrate ions from the overuse of nitrogen-based fertilizers and sewage discharge severely disturb the natural nitrogen cycle[1–3]. Among the reported nitrate removal technologies, renewable energy-driven electrocatalytic nitrate reduction to ammonia (NRA) is promising due to the utilization of water as the hydrogen source and green electrons as reductants[4–8]. Various strategies have been developed to enhance the NRA performance, and most of the reported catalysts have shown high Faradaic efficiency and ammonia yield rate for high-concentration (>100 mM) nitrate reduction[9–15]. However, the efficient electroreduction of low-concentration nitrate (≤10 mM) is still a major challenge, mainly due to the restricted migration of nitrate ions near the working electrode and the presence of competitive hydrogen evolution reaction[16–18]. Thus, developing an alternative strategy to break the mass-transfer limitation near the cathode surface and retard the competitive reaction is of great significance to achieving efficient low-concentration nitrate electroreduction to ammonia, which is crucial for the practical application of the NRA.

The pulse electrolysis approach, in which the applied potential/current changes periodically, has been widely used in hydrogen peroxide production[19], organic electrosynthesis[20], and carbon dioxide

[1]Institute of Molecular Plus, School of Science, Tianjin University, 300072 Tianjin, China. [2]Haihe Laboratory of Sustainable Chemical Transformations, 300192 Tianjin, China. [3]School of Materials Science and Engineering, University of Science and Technology Beijing, 100083 Beijing, China. [4]Institute of New Energy Materials, School of Materials Science and Engineering, Tianjin University, 300072 Tianjin, China. [5]Shunde Innovation School, University of Science and Technology Beijing, 528399 Foshan, China. [6]Tianjin University-Asia Silicon Joint Research Center of Ammonia-Hydrogen New Energy, 810000 Xining, China. [7]These authors contributed equally: Yanmei Huang, Caihong He. ✉e-mail: qipeng@ustb.edu.cn; yyu@tju.edu.cn

reduction[21–23] due to its intrinsic advantages in modulating the local microenvironments that are inaccessible under conventional potentiostatic/galvanostatic conditions[24–26]. Recently, Li et al. reported a tandem catalysis process for NRA, in which two alternated negative potentials were used for the cascade conversion of nitrate to nitrite and nitrite to ammonia, respectively[27]. In spite of this, it is worth noting that the periodic appearance of a positive voltage is expected to replenish the negatively charged $NO_3^-$ in the vicinity of the working electrode and suppress the competitive hydrogen evolution reaction. However, the corresponding studies are still lacking, leading to an obscure understanding of the specific mechanism. Thus, we adopt a periodically alternated positive and negative potential to promote low-concentration nitrate reduction to ammonia. Intermetallic nanocrystals with atomic-level definite structures are a model platform for structure-performance relationship investigations[28–31]. Ruthenium (Ru) has been reported as a highly active species for NRA[7,32,33]. Incorporating indium (In) as the promoter can further optimize the electronic structure of the catalyst and decrease the use of the noble metal Ru[16,34].

In this work, carbon-supported $RuIn_3$ intermetallic compounds ($RuIn_3$/C) are chosen as model catalysts for pulsed electroreduction of low-concentration nitrate (≤10 Mm) to ammonia. A high ammonia Faradaic efficiency (97.6%) and yield rate (2.7 mmol⁻¹ h⁻¹ mg$_{Ru}$⁻¹) are achieved over $RuIn_3$/C under pulsed conditions, surpassing those obtained under potentiostatic conditions (Faradaic efficiency: 65.8%, yield rate: 1.0 mmol⁻¹ h⁻¹ mg$_{Ru}$⁻¹). The combined results of in situ Raman, in situ attenuated total reflection Fourier transform infrared (ATR-FTIR) spectroscopy, online differential electrochemical mass spectrometry (DEMS), and finite element analysis (FEA) demonstrate that the periodic appearance of positive potential can significantly

increase the $NO_3^-$ ion concentration near the working electrode and optimize the adsorption configuration of the key *NO intermediate, which synergistically promote the reduction of low-concentration nitrate to ammonia.

## Results

### Synthesis and characterizations of $RuIn_3$ intermetallic catalysts

$RuIn_3$/C is prepared through a solid-phase grinding method followed by an $H_2$ annealing treatment. The X-ray diffraction (XRD) pattern (Fig. 1a) of the prepared sample is consistent with the $RuIn_3$ intermetallic compounds with a typical tetragonal crystal structure (ICDD No.04-004-7014), which belongs to the space group of P42/mnm (a = b = 6.995 Å, c = 7.236 Å). The corresponding Rietveld refinement results demonstrate the high phase purity of the obtained material (Fig. 1a and Supplementary Table 1). Representative high-angle annular dark-field scanning transmission electron microscopy (HAADF-STEM) images (Fig. 1b and Supplementary Fig. 1) show the small nanoparticles of $RuIn_3$ with an average size of ~6.1 nm deposited on the carbon black support. The atomic-resolution HAADF-STEM image reveals the highly ordered distribution of Ru and In atoms throughout the whole nanoparticle along the [$\bar{2}$12] zone axis (Fig. 1c). Moreover, the alternating arrangements of Ru and In atoms are in good agreement with the simulated model (Fig. 1d, e). The corresponding fast Fourier transform (FFT) pattern of $RuIn_3$ shows that the obtained nanocrystal is a single crystal with the characteristic diffraction pattern along the [$\bar{2}$12] zone (Fig. 1f), which is consistent with the simulated diffraction pattern (Fig. 1g). The diffraction streaks, including (101), (120), and ($\bar{1}$2$\bar{2}$), confirm the high degree of atomic ordering. The energy-dispersive X-ray spectroscopy (EDX) elemental mapping images (Fig. 1h) of an

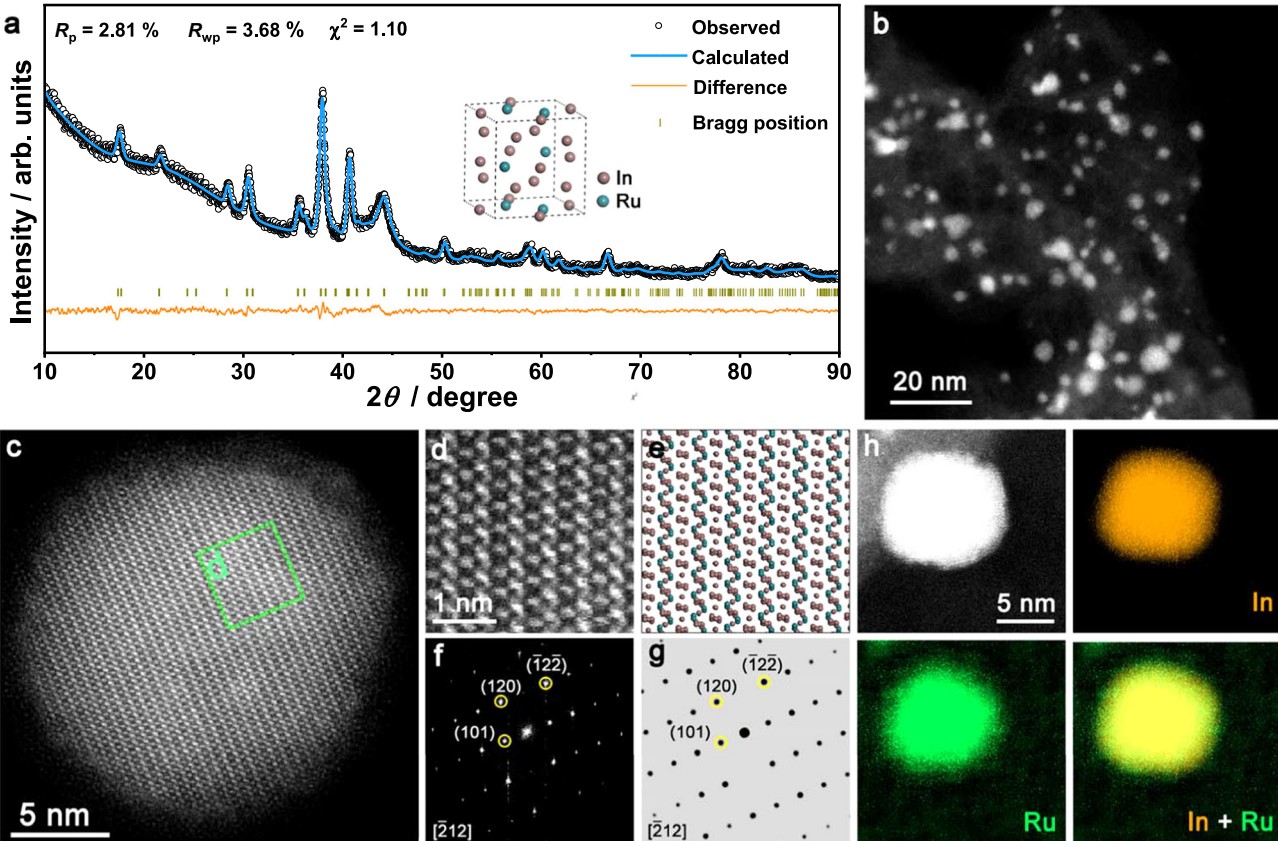

**Fig. 1 | Characterizations of $RuIn_3$/C. a** XRD pattern and the Rietveld refinement of $RuIn_3$/C, inset: the crystal structure model of $RuIn_3$/C. **b** HAADF-STEM image of $RuIn_3$/C. **c** Atomic-resolution HAADF-STEM image of $RuIn_3$/C along the [$\bar{2}$12] zone axis. **d** The magnified image and **e** the corresponding crystal structure model of the dashed square region in (**c**). **f** The FFT pattern and **g** the corresponding simulated diffraction patterns of $RuIn_3$/C along the [$\bar{2}$12] zone axis. **h** HAADF-STEM image of $RuIn_3$ nanoparticle and the corresponding EDX elemental mapping images.

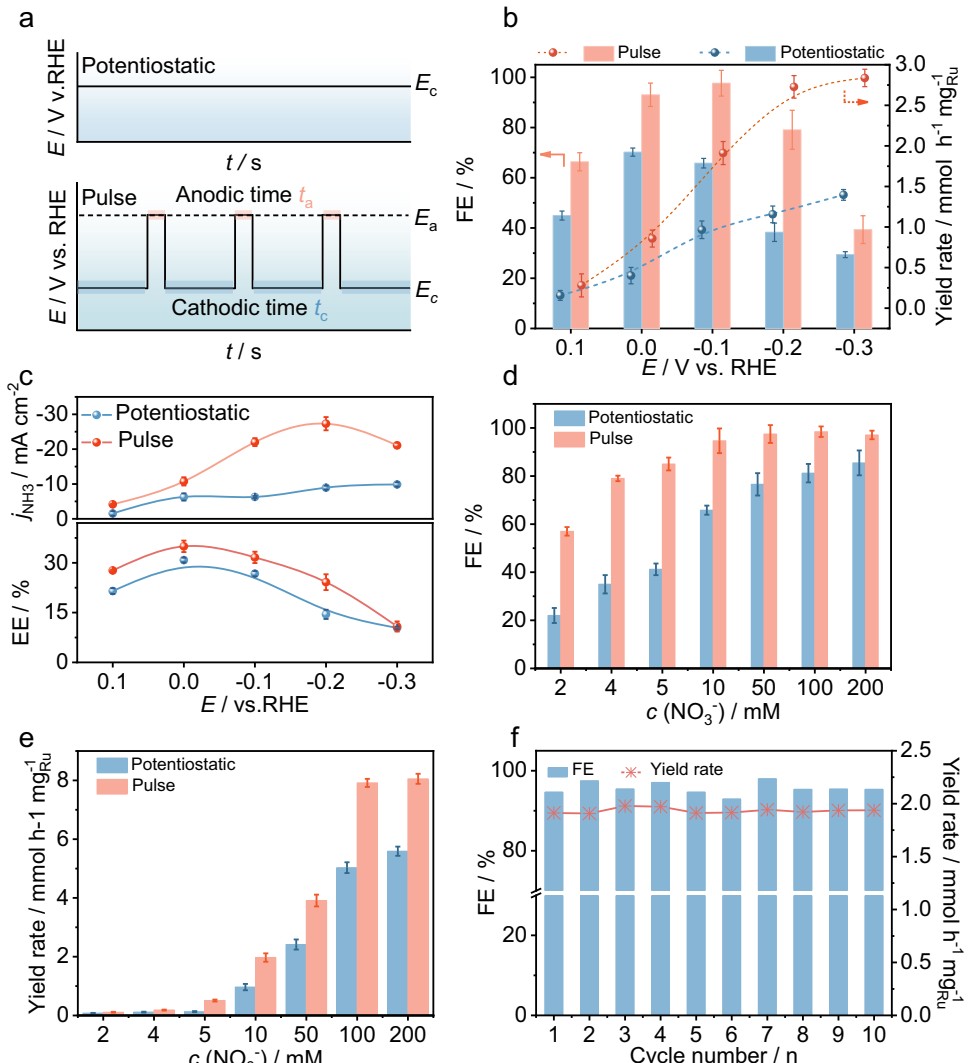

**Fig. 2 | Pulsed NRA performance of RuIn₃/C. a** Schematic illustration of potentiostatic and pulsed potential tests. The NRA performance comparison over RuIn₃/C between potentiostatic and pulsed conditions: **b** ammonia Faradaic efficiency and yield rate, **c** ammonia partial current density and energy efficiency. Note: the parameters applied in pulsed condition are $E_a = +0.6\,V$, $t_c = 4\,s$, $t_a = 0.5\,s$. **d** Faradaic efficiency and **e** yield rate of NRA over RuIn₃/C under potentiostatic ($E = -0.1\,V$) and pulsed conditions ($E_c = -0.1\,V$, $E_a = +0.6\,V$, $t_c = 4\,s$, $t_a = 0.5\,s$) with different nitrate concentrations. **f** Cyclic durability test of RuIn₃/C for 10 Mm nitrate electroreduction to ammonia under pulsed conditions ($E_c = -0.1\,V$, $E_a = +0.6\,V$, $t_c = 4\,s$, $t_a = 0.5\,s$).

individual RuIn₃ nanoparticle display the homogenous distribution of both elements without phase segregation. The X-ray photoelectron spectroscopy (XPS) spectra (Supplementary Fig. 2) show that Ru mainly stays in the metallic state, while the majority of In is in the oxidized state, which can be ascribed to the surface oxidation of low-coordinated superficial indium atoms when exposed to air[30,35]. The atomic ratio of Ru to In is 74.1:25.9 (Supplementary Fig. 3), consistent with the theoretical stoichiometric ratio and the result obtained from inductively coupled plasma optical emission spectroscopy (ICP-OES) (Supplementary Table 2). In addition, carbon-supported Ru nanoparticles (Ru/C) and carbon-supported In nanoparticles (In/C) are also prepared for comparison using the same synthetic methods as RuIn₃/C (Supplementary Fig. 4).

## Pulsed NRA performance over RuIn₃/C catalysts

The electrochemical NRA performance of RuIn₃/C is investigated under ambient conditions in a standard three-electrode H-cell using 0.1 M KOH and 10 Mm NO₃⁻ as the electrolyte. Ion concentrations are determined by ultraviolet-visible spectrometer (Supplementary Fig. 5). Unless otherwise stated, all potentials are corrected versus the

reversible hydrogen electrode (RHE). The linear scan voltammetry (LSV) curves show a noticeable increase in current density after adding 10 mM nitrate, indicating the efficient electroreduction of nitrate by RuIn₃/C (Supplementary Fig. 6). Figure 2a illustrates the typical pulse sequence: NRA occurs at cathodic potential ($E_c$) for $t_c$ s followed by anodic potential ($E_a$) for $t_a$ s, and then the alternating change in potential repeats. The related parameters for pulsed NRA are systematically screened (Supplementary Figs. 7–9) as well. With the ammonia Faradaic efficiency and yield rate as evaluating indicators, the optimal parameters of $E_a = 0.6\,V$, $E_c = -0.1\,V$, $t_a = 0.5\,s$, and $t_c = 4\,s$ are selected and used for the following experiments. Compared with potentiostatic conditions, pulsed conditions exhibit much higher performance. Specifically, the ammonia Faradaic efficiency increases by ~40% (potentiostatic: 65.8%, pulsed: 97.6%) at $E_c = -0.1\,V$ (Fig. 2b). At $E_c = -0.2\,V$, the pulsed NRA shows a ~2.4-fold increase in the highest ammonia yield rate (potentiostatic: 1.2 mmol⁻¹ h⁻¹ mg_Ru⁻¹, pulsed: 2.7 mmol⁻¹ h⁻¹ mg_Ru⁻¹) and a 3.1-fold increase in NH₃ partial current density (potentiostatic: 8.9 mA cm⁻², pulsed: 27.3 mA cm⁻²) (Fig. 2b, c). The ammonia energy efficiencies show an optimal value (~35%) at $E_a = +0.6\,V$, $E_c = 0\,V$, $t_c = 4\,s$, $t_a = 0.5\,s$ (Fig. 2c). Moreover, D-isotope and

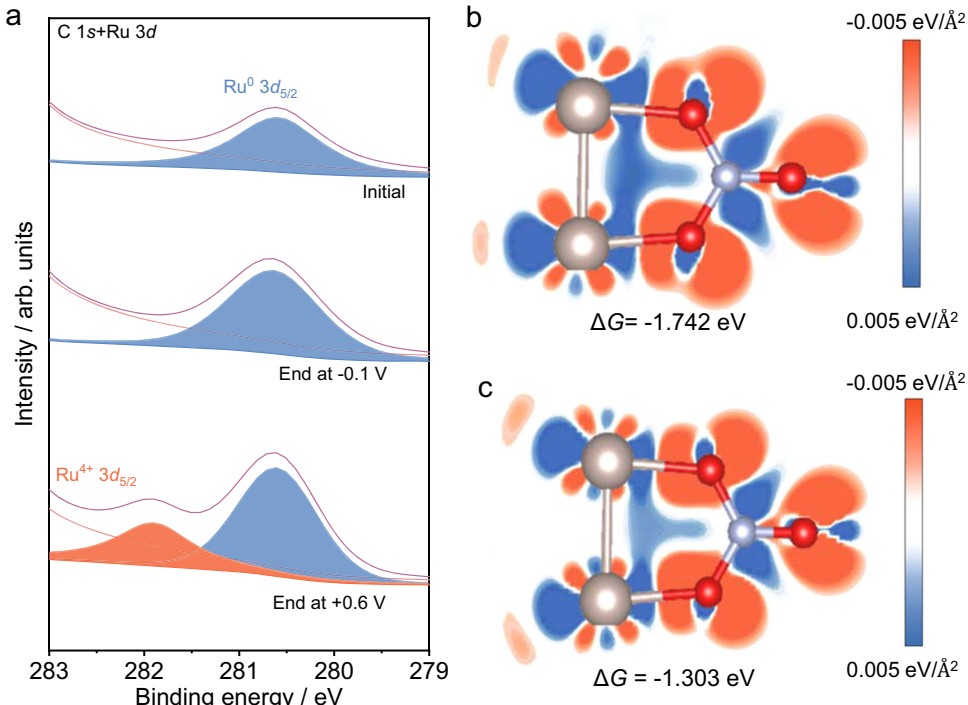

**Fig. 3 | Active species analysis and DFT calculations. a** Ru 3$d$ XPS spectra of RuIn$_3$/C before pulse electrolysis and ending the sequence at different potentials. The difference charge density of NO$_3^-$ over Ru sites on the surface of **b** RuIn$_3$/C and **c** Ru/C. Gray, red, and blue spheres represent the Ru, N, and O atoms. The blue electronic cloud indicates charge depletion, and the red electronic cloud indicates charge accumulation.

$^{15}$N-isotope labeling experiments prove that hydrogen and nitrogen in the produced NH$_4^+$ originate from H$_2$O and nitrate, respectively (Supplementary Fig. 10). In addition, the electric energy consumption and production cost of 1 kg NH$_3$ over RuIn$_3$/C through pulsed electrolysis are calculated as ~20.7 kWh and US$0.62 based on the price of renewable electricity (US$0.03 kWh$^{-1}$) (Supplementary Fig. 11 and Supplementary Table S3)[36,37]. Although this assessment only considers electricity costs and excludes capital expenses and ohmic losses, the reported RuIn$_3$/C for pulsed NRA remains appealing due to the environmental advantages of nitrate contaminant removal. When the nitrate concentrations vary from 2 mM to 200 mM, the pulsed NRA performance is maintained well (Fig. 2d, e). The enhancement is more prominent when the NO$_3^-$ concentration is lower, indicating the advantage of pulsed electrolysis in treating low-concentration nitrate. Simultaneously, the ammonia Faradaic efficiency and yield rate remain well throughout the ten-cycle stability tests (Fig. 2f and Supplementary Fig. 12a). There are no obvious changes in the XRD patterns, pH values, TEM image, and the particle size distribution of the used RuIn$_3$/C samples (Supplementary Fig. 12b–e). These results demonstrate the stability of RuIn$_3$/C catalysts. Furthermore, the nitrate concentration can be continuously reduced to 0.35 mM under pulsed conditions (Supplementary Fig. 13), which is lower than the emission standard (≤0.81 mM)[38]. For potentiostatic conditions, the residual concentration is far from the emission standard (Supplementary Fig. 13). The calculated double-layer capacitances before and after the pulsed tests are 5.63 and 5.95 mF cm$^{-2}$, respectively (Supplementary Fig. 14), indicating that the surface reconstruction caused by the pulsed potential can be ignored.

## Active species analysis of pulsed NRA over RuIn$_3$/C
The same XRD patterns of RuIn$_3$/C ending at different potentials after pulsed potential tests and potentiostatic tests suggest the maintenance of the bulk intermetallic structure during performance tests (Supplementary Fig. 15). Compared to RuIn$_3$/C and Ru/C, In/C shows

ignorable activity for NRA under the same conditions (Supplementary Figs. 6 and 16), indicating the active species role of ruthenium. After normalization to the electrochemical active surface area (ECSA), RuIn$_3$/C still shows a higher current density than Ru/C, proving its high intrinsic activity (Supplementary Fig. 17). XPS spectra (Fig. 3a) and in situ Raman spectra (Supplementary Fig. 18) show that the dominant ruthenium species on the RuIn$_3$/C surface is Ru$^0$ at −0.1 V ($E_c$) and Ru$^{4+}$ at +0.6 V ($E_a$), respectively. In addition, RuIn$_3$/C and Ru/C exhibit no NRA performance under +0.6 V (Supplementary Fig. 19). Thus, the active species during the NRA process lie in metallic Ru$^0$ species. The surface indium mainly exists in the oxide state before and after electrolysis (Supplementary Figs. 20 and 21), which can be ascribed to the unfavorable thermodynamics of In$_2$O$_3$/In conversion ($\varphi^{\ominus}_{(In_2O_3/In)}$ = −1.034 V vs. SHE[35]. Subsequently, a theoretical simulation is performed to interpret the role of surface In$_2$O$_3$ (Fig. 3b, c). The electron cloud between NO$_3^-$ and Ru sites is larger in the presence of In$_2$O$_3$, indicating a stronger electron interaction. The calculated absorption free energy of nitrate on the metallic Ru$^0$ species with In$_2$O$_3$ (−1.742 eV) is lower than that without In$_2$O$_3$ (−1.303 eV), proving the positive effect of In$_2$O$_3$ on the adsorption of nitrate over Ru sites.

Subsequently, in situ and ex situ experiments are carried out to understand the pulsed electrolysis process. The variations of the NO$_3^-$ concentration at the electrode surface under different potentials are recorded via in situ Raman spectroscopy. Under −0.1 V, the characteristic peak of nitrate (~1050 cm$^{-1}$)[9] gradually shrinks with the accumulation of scanning times and almost vanishes after 125 s of reduction (Fig. 4a and Supplementary Fig. 22a). As the potential is switched to +0.6 V, the peak at ~1050 cm$^{-1}$ emerges again, demonstrating nitrate accumulation near the positively charged electrode. In addition, potential-dependent in situ Raman experiments using 0.1 M KOH + 10 mM NH$_4$Cl as the electrolyte exclude the possibility that the accumulated NO$_3^-$ originated from NH$_4^+$ oxidation (Supplementary Fig. 22b). When the applied potential switched between −0.1 V and +0.6 V periodically, the normalized peak intensity of nitrate

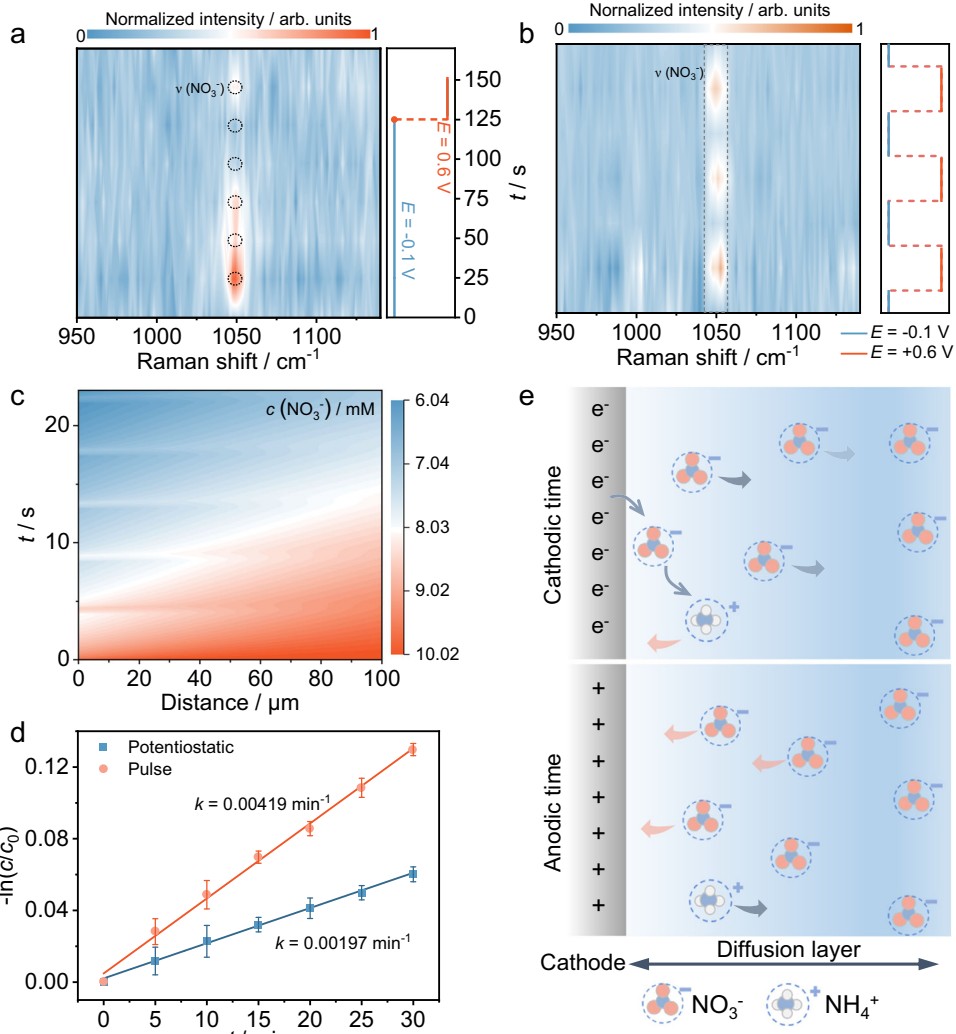

**Fig. 4 | Kinetics analysis during pulsed NRA. a** In situ Raman spectra of RuIn$_3$/C under −0.1 V for 125 s and then +0.6 V in 0.1 M KOH + 10 mM NO$_3^-$ solution. **b** In situ Raman spectra of RuIn$_3$/C under −0.1 V and then +0.6 V in 0.1 M KOH + 10 mM NO$_3^-$ solution with 3 cycles. **c** The FEA simulated NO$_3^-$ distribution at the cathode- solution interface under pulsed conditions with an initial nitrate concentration of 10 mM. **d** The linearized pseudo-first-order kinetic profiles under potentiostatic and pulsed conditions. **e** Schematic illustration for the effect of pulsed voltage on the mass transport of different species.

(-1050 cm$^{-1}$) exhibited a nearly periodic variation trend, in which the characteristic peak displays a higher intensity at +0.6 V and then decreases to a relatively lower level at −0.1 V. (Fig. 4b and Supplementary Fig. 22c). Moreover, the nitrate accumulation effects under different $E_a$ values are also considered. The peak intensity of nitrate at +0.6 V is almost the same as that at +0.8 V, indicating that +0.6 V is sufficient to renew the nitrate distribution near the electrode surface (Supplementary Fig. 23). Since the concentration profiles of nitrate near the cathode surface at the second scale are difficult to measure, FEA simulations are conducted for potentiostatic and pulsed NRA. Under a constant potential of −0.1 V, interfacial NO$_3^-$ is depleted soon (Supplementary Figs. 24 and 25). While under the pulsed potential, NO$_3^-$ is resupplied during the anodic time (Fig. 4c and Supplementary Fig. 25). As a result, the NO$_3^-$ concentration at a specific time and distance in the diffusion layer under pulsed conditions is higher than that under potentiostatic conditions. A pseudo-first-order rate law is obtained according to the time-dependent nitrate concentration changes in electrolytes (Supplementary Fig. 26) under different conditions. The apparent rate constants are calculated to be 0.00197 min$^{-1}$ with a constant potential and 0.00491 min$^{-1}$ under pulsed conditions (Fig. 4d). This indicates that manipulating interfacial ion diffusion by the pulsed voltage can improve reaction kinetics. According to the

results above, the effect of pulsed voltage on the mass transport of different species is illustrated in Fig. 4e. During the cathodic time, NO$_3^-$ is transferred to the cathode by diffusion, absorbed on the electrode surface, and finally reduced to ammonia by electrons from the electrode. A pulsed voltage with periodic anodic potential can mitigate the mass transport limitation by replenishing NO$_3^-$ in the diffusion layer while the generated NH$_4^+$ diffuses from the interface to the bulk solution. Timely replenishment of NO$_3^-$ at the diffusion layer is beneficial to break the dynamic limit aroused by the mass transport and finally enhance the catalytic performance. Moreover, rotating disk electrode (RDE) experiments are carried out to better understand the pulse function (Supplementary Fig. 27). Compared with the non-stir potentiostatic condition, the ammonia yield rate increases by ~60% than at the stir potentiostatic condition (1800 rpm), indicating the existence of mass-transfer limitation for low-concentration nitrate electroreduction. Notably, the ammonia yield rate further increases under pulse conditions without stirring. These results prove that the pulse approach can not only weaken the mass-transfer limitation of low-concentration nitrate reduction but also promote the thermodynamics of NRA.

A series of in situ and ex situ characterizations are performed to understand the reaction pathway during pulsed NRA. Online DEMS is

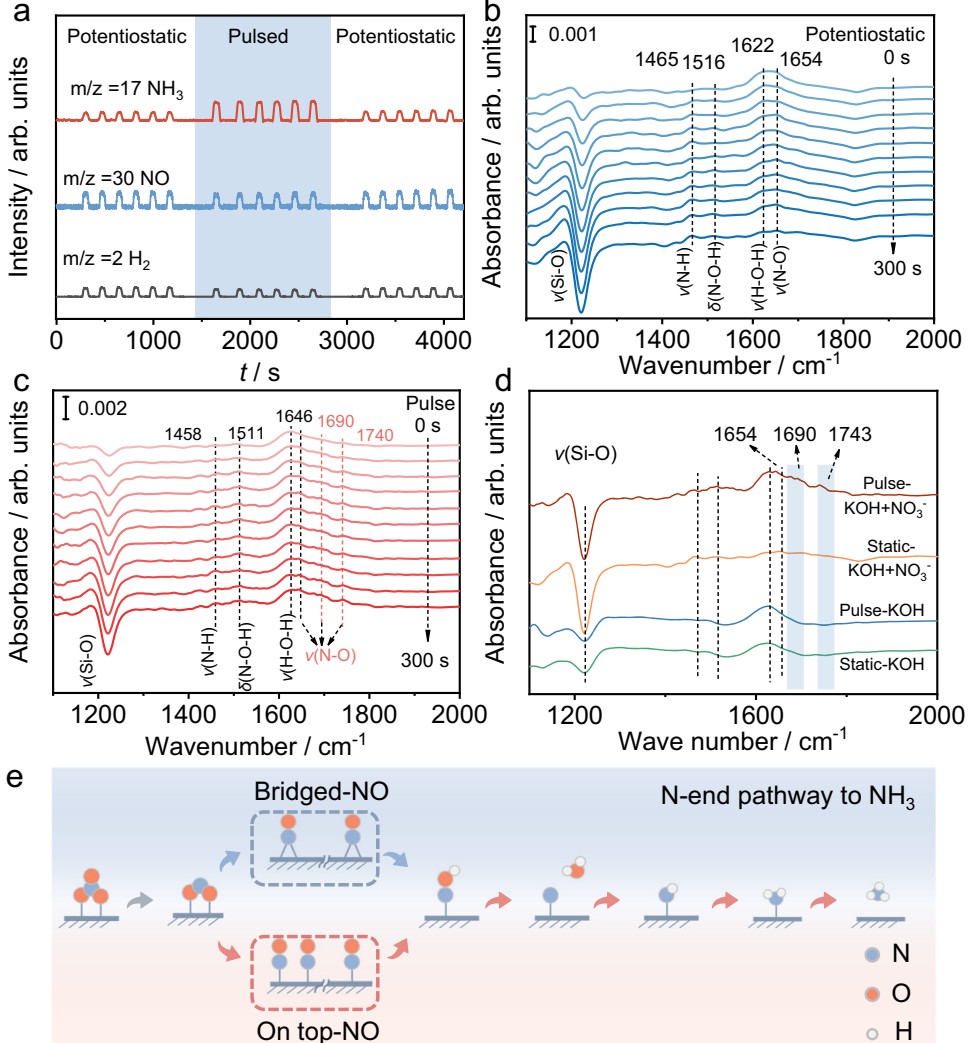

**Fig. 5 | Reaction pathway analysis during pulsed NRA. a** Online DEMS during the NRA process, starting with a constant potential ($E = -0.1$ V), then switching to a pulsed potential ($E_c = -0.1$ V, $E_a = +0.6$ V, $t_c = 4$ s, $t_a = 0.5$ s), and finally switching back to the starting constant potential. Time-dependent in situ ATR-FTIR spectra of NRA under **b** constant potential ($E = -0.1$ V), **c** pulsed potential ($E_c = -0.1$ V, $E_a = +0.6$ V, $t_c = 4$ s, $t_a = 0.5$ s). **d** The comparison of in situ ATR-FTIR spectra under the different conditions at $t = 300$ s. **e** Schematic illustration for the deduced NRA reaction pathway over RuIn$_3$/C.

carried out to capture the volatile intermediates and products (Supplementary Fig. 28). Along with time extension, the signals of m/z 2, 17, 28, 30, and 44 are detected (Fig. 5a and Supplementary Fig. 29), which could be assigned to H$_2$, NH$_3$, N$_2$, NO, and N$_2$O[39]. During the pulsed potential process, the signal intensities of H$_2$, N$_2$, NO, and N$_2$O shrink while the NH$_3$ intensity increases (Supplementary Fig. 29). Moreover, the trend is the opposite when the power mode shifts back to a constant potential. Thus, the periodic change in signal intensities demonstrates the promotion of NH$_3$ generation and the suppression of H$_2$ during pulsed NRA. Subsequently, electrochemical in situ ATR-FTIR is applied to monitor the absorbed intermediates. The peak intensity is expressed in absorbance, and the positive peak represents product generation. Compared with the in situ ATR-FTIR experiment performed in 0.1 M KOH (Supplementary Fig. 30), the bands assigned to ammonia production ($-1460$ cm$^{-1}$)[40], bending vibration of NOH ($-1510$ cm$^{-1}$), and bridged NO ($-1650$ cm$^{-1}$)[41-43] can be observed under both potentiostatic and pulsed conditions. Their intensity increases with the proceeding of NRA (Fig. 5b, c). Notably, the extra bands at $-1690$ cm$^{-1}$ and $-1740$ cm$^{-1}$ (designated to on-top adsorption of NO[44,45]) can only be observed under pulsed conditions (Fig. 5c, d). Based on the information offered by online DEMS, in situ ATR-FTIR, and reported

literature[32,46], the possible reaction pathways are deduced (*NO$_3$ → *NO$_2$ → *NO → *NOH → *N → *HN → *H$_2$N → *NH$_3$) (Fig. 5e). It is well known that NO is considered the critical intermediate in NRA[47,48]. Furthermore, it has been reported that the reduction rate of on-top adsorbed NO is higher than that of bridged adsorbed NO[43,45,49]. Thus, the pulsed potential approach promotes the formation of a more active NO absorption configuration, enhancing NRA performance.

## Discussion

This study systematically demonstrates the advantages of pulsed potential for enhancing electrocatalytic reduction of low-concentration nitrate to ammonia performance. An optimized pulsed potential sequence ($E_c = -0.1$ V, $t_c = 4$ s, $E_a = 0.6$ V, $t_a = 0.5$ s) is introduced for NRA using RuIn$_3$/C as a model platform. Pulsed electrolysis endows an excellent performance for converting low-concentration nitrate (Faradaic efficiency: 97.6% yield rate: 2.7 mmol$^{-1}$ h$^{-1}$ mg$_{Ru}^{-1}$ for 10 mM nitrate), much superior to the other reported catalysts under potentiostatic conditions. Meanwhile, the nitrate concentration can be continuously reduced to 0.35 mM under pulsed conditions, lower than the emission standard ($\leq 0.81$ mM). The combined results of in situ Raman and XPS prove that the active

species in RuIn$_3$/C under both potentiostatic and pulsed conditions lie in Ru$^0$. The combined results of mass-transfer analysis (in situ Raman and FEA) and intermediate detection (online DEMS and in situ ATR-FTIR) reveal that pulsed electrolysis can promote the accumulation of nitrate near the electrode and favor the formation of a more active *NO intermediate with on-top absorption configuration, which synergistically enhances the reduction of low-concentration nitrate to ammonia. The proposed pulsed electrolysis technology has implications beyond NRA conversion, as favorability could be tuned for other competing electrochemical reactions.

## Methods

### Synthesis of RuIn$_3$/C

In a typical synthesis, a mixture of 0.025 mmol RuCl$_3$·3H$_2$O and 0.075 mmol In(NO$_3$)$_3$·xH$_2$O, 0.30 g CH$_2$N$_2$, and 22 mg of carbon support (Ketjen Black EC300J) were mixed and grinded using an agate mortar and pestle. After sufficient grinding, the mixture was transferred into a porcelain boat and placed into a tube furnace for heat treatment. The whole process of heating was carried out under an atmosphere of 5% H$_2$/Ar. First, the mixture was heated from room temperature to 180 °C at 5 °C min$^{-1}$ and kept for 30 min. Subsequently, the temperature was increased to 900 °C at 5 °C min$^{-1}$ and maintained for 12 h. After cooling to room temperature, the obtained product was treated with 0.5 M H$_2$SO$_4$ to remove the oxides. In addition, carbon-supported Ru nanoparticles (Ru/C) and carbon-supported In nanoparticles (In/C) were also prepared for comparison using the same synthetic methods as RuIn$_3$/C.

### Characterization

The crystal structure of the prepared materials was characterized by XRD. XRD patterns were collected on a Bruker D8 X-ray diffractometer operating at 40 kV and 40 mA with Cu Kα radiation (λ = 0.154 nm) and recorded with 2θ ranging from 10° to 90° at a sweep speed of 10° min$^{-1}$. The morphology of the prepared samples was characterized by TEM (JEOL JEM-2100F, 200 kV). Atomic-level characterizations of the prepared samples, such as aberration-corrected HAADF-STEM images and elemental mapping images, were obtained by an aberration-corrected JEOL JEM-ARM300F Grand ARM transmission electron microscope operated at an accelerating voltage of 300 kV equipped with a cold field-emission electron gun and SDD-type EDX detectors. The surface elemental analysis of the as-prepared nanomaterials was conducted by XPS on a Thermo Scientific ESCALAB 250Xi spectrometer using Al Kα radiation as the X-ray source for excitation. The C 1s peak energy at 284.8 eV was used for calibration. ICP-OES was performed with an Optima 7300 DV. The UV–vis absorbance spectra were measured on a Beijing Purkinje General T6 new century spectrophotometer.

### Electrochemical measurements

The electrochemical performance tests were carried out on a CHI 650E electrochemical workstation using a three-electrode configuration within an H-cell. Here, 40 mL of 0.1 M KOH (containing 10 mM NO$_3^-$) solution was added into the cathode compartment, and another 40 mL of 0.1 M KOH solution was added into the anode compartment. Then, 1 mg of the as-prepared catalyst and 20 μL of 5 wt.% Nafion solution was dispersed in 980 μL ethanol by ultrasonication for 20 min to obtain a uniform suspension. Then, 200 μL of the suspension was dropped onto a carbon paper electrode with a controlled area of 1 × 1 cm$^2$ and used as the working electrode (WE). Platinum foil and a Hg/HgO (filled with 0.1 M KOH) acted as the counter electrode (CE) and reference electrode (RE), respectively. Unless otherwise stated, all the provided potentials were converted to the reversible hydrogen electrode (RHE) scale ($E_{RHE} = E_{Hg/HgO} + 0.0591 \times pH + 0.164$). The current density reported in this work is based on the geometric surface area. LSV experiments were conducted at a scan rate of 10 mV s$^{-1}$. The cyclic voltammograms were recorded at a scan rate of 10 mV s$^{-1}$ in 0.1 M KOH

solution. The potentiostatic measurement was performed from 0.1 V to −0.3 V vs. RHE for 1 h. The total time for pulse electrolysis, including cathodic and anodic durations, was controlled to be 1 h. Error bars correspond to the standard deviations of three independent measurements, and the center value for the error bars is the average of the three independent measurements. A UV–vis spectrophotometer was used to detect the ion concentration of pre-and post-test electrolytes after diluting to the appropriate concentration to match the range of calibration curves. The concentrations of nitrate, nitrite, and ammonia ions were determined by UV-vis spectrophotometric method and calculated by the corresponding standard calibration curve[50,51]. The ECSA was assessed through the double-layer capacitance method in 0.1 M KOH solution within the non-Faradic potential range, employing various scan rates ranging from 5 to 100 mV s$^{-1}$. The ECSA of the working electrodes was calculated according to the following equations (Eqs. (1) and (2)):

$$I_c = v \times C_{dl} \tag{1}$$

$$ECSA = \frac{C_{dl}}{C_s} \tag{2}$$

The $I_c$ represents the charging current with different scan rates, $v$ is the scan rate, and $C_{dl}$ is the double-layer capacitance. $C_s$ represents the specific capacitance for a flat metallic surface, generally during the 20−60 μF cm$^{-2}$ range.

### Performance analysis

The calculation method of ammonia Faraday efficiencies in the pulsed NRA experiment was referenced in the literature[21,22,27,52,53] and listed below (Eq. (3)):

$$FE = \frac{c \times V \times n \times F}{Q} = \frac{c \times V \times n \times F}{\int_0^t I dt} \tag{3}$$

where $c$ is the product concentration; $V$ is the volume of electrolyte; $n$ is the number of transferred electrons; $F$ is the Faraday constant; Q is obtained from the integral of the $I$-$t$ curve and is the total charge passed through the electrode.

The ammonia yield rate was calculated as follows (Eq. (4)):

$$Yield\ rate = \frac{c \times V}{m_{cat} \times M \times t} \tag{4}$$

where $c$ is the product concentration; $V$ is the volume of electrolyte; $m_{cat}$ is the mass of catalysts; $t$ is the total electrolysis duration; and $M$ is the molar mass of NH$_3$.

### In situ Raman spectroscopy

In situ Raman measurements were carried out on a Renishaw inVia reflex Raman microscope under an excitation of 532 nm laser light with 50 mW. A homemade cell and CHI650E electrochemical workstation were used. The cell was made of Teflon and equipped with a quartz glass plate to protect the objective. The WE (carbon paper coated with catalysts) was submerged in the electrolyte and kept perpendicular to the laser. Hg/HgO (filled with fresh 0.1 M KOH) acted as the RE. A platinum wire embraced around the cell wall served as the CE. It took 25 s to complete one scan.

### Online differential electrochemical mass spectrometry (DEMS) measurement

The online DEMS measurements were performed on a QAS 100 (Linglu Instruments (Shanghai) Co. Ltd) to capture the volatile intermediates and products. For the online DEMS test, carbon paper coated with catalysts, platinum wire, and Hg/HgO (filled with fresh

0.1 M KOH) electrodes were employed as the WE, CE, and RE, respectively. Here, 0.1 M KOH containing 0.01 M $NO_3^-$ kept flowing through the homemade electrochemical cell through a peristaltic pump. The signal was gathered using a hydrophobic polytetrafluoroethylene (PTFE) membrane, essential for allowing volatile compounds while preventing water entry into the vacuum chamber. The produced volatile products were brought to the mass spectrometer through a pump. The DEMS tests include two modes. The first mode is the continuous power-on mode with the application of continuous constant potential ($E = −0.1$ V) or pulsed potential ($E_c = −0.1$ V, $E_a = +0.6$ V, $t_c = 4$ s, $t_a = 0.5$ s). The second mode is the alternative power switch on and off. The second mode test includes three stages. In the first stage, the switch-on potential is constant ($E = −0.1$ V), and the switch on and off repeats 6 times. In the second stage, the switch-on potential is pulsed ($E_c = −0.1$ V, $E_a = +0.6$ V, $t_c = 4$ s, $t_a = 0.5$ s), and the process of switching on and off repeats 6 times. In the third stage, the switch-on potential is constant ($E = −0.1$ V), and the switch on and off repeats 6 times. The duration time for switch on and off are 50 s and 180 s, respectively.

### In situ attenuated total reflection Fourier transformed infrared spectroscopy (ATR-FTIR) measurement

In situ ATR-FTIR was conducted on Nicolet Nexus 670 Spectroscopy system using a Linglu Instruments ECIR-II cell coupled with a Pike Veemax III ATR accessory. The monocrystal silicon was initially coated with an Au layer using the chemical plating method as reported[7] without modification to improve the signal intensity. Then, 1 mg of catalyst and 20 μL of 5 wt.% Nafion solution were added to 980 μL of absolute ethanol and ultrasonicated for 20 min to obtain a homogenous ink. Next, 60 μL of ink was carefully dropped on the surface of the gold film and used as the WE. Platinum foil and Hg/HgO (filled with fresh 0.1 M KOH) electrodes served as the CE and RE, respectively. Then, 0.1 M KOH with and without 0.01 M $NO_3^-$, the solution was employed as the electrolyte, respectively. Time-dependent spectra were collected at a constant potential (−0.1 V) and pulsed potential ($E_c = −0.1$ V, $E_a = +0.6$ V, $t_c = 4$ s, $t_a = 0.5$ s).

### DFT calculations

All DFT calculations were performed using the Vienna Ab initio Simulation Package (VASP)[54]. The projector augmented wave (PAW)[55] pseudopotential with the SPBE[56] generalized gradient approximation (GGA) exchange-correlation function was utilized in the computations. The cutoff energy of the plane waves basis set was 500 eV, and a Monkhorst-Pack mesh of $3 \times 3 \times 1$ was used in K-sampling. The long-range dispersion interaction was described by the DFT-D3 method. The electrolyte was incorporated implicitly with the Poisson-Boltzmann model implemented in VASPsol[57]. The relative permittivity of the media was chosen as $\epsilon_r = 78.4$, corresponding to that of water. All structures were spin-polarized, and all atoms were fully relaxed with the energy convergence tolerance of $10^{-5}$ eV per atom, and the final force on each atom was $<0.05$ eV Å$^{-1}$. All periodic slabs have a vacuum layer of at least 15 Å. All of the atoms could relax during geometry optimization. The adsorption energy of reaction intermediates can be computed using the following equation: $\triangle E_{ads} = E_{*ads} - E_* - E_{ads}$, where $E_{*ads}$, $E_*$, and $E_{ads}$ denote the total energy of the adsorbed system, the clear surface and a single $NO_3^-$ group, respectively.

### Finite element analysis (FEA)

The transport of the species in the electrolyte is governed by the Nernst-Planck equations (Eq. (5))

$$N_i = -D_i \nabla c_i + \frac{z_i F}{RT} D_i c_i \nabla \emptyset_L \qquad (5)$$

The material balances were expressed through (Eq. (6))

$$\frac{\partial c_i}{\partial t} + \nabla \cdot N_i = R_{i,\text{tot}} \qquad (6)$$

The electroneutrality condition is invoked by the following expression (Eq. (7))

$$\sum_i z_i c_i = 0 \qquad (7)$$

Where the reaction rate $R_i$ on the electrode is derived by Faraday's Law of Electrolysis, $N_i$ is the total flux of species i, the flux in an electrolyte is described by the Nernst–Planck equations and accounts for the flux of charged solute species (ions) by diffusion and migration in this system, $c_i$ represents the concentration of ion i, $z_i$ is its valence, $D_i$ is the diffusion coefficient, F denotes the Faraday constant, the electrolyte potential, and $\phi_L$ is the ionic potential of the electrolyte. At the bulk boundary, a uniform concentration was assumed to be equal to the bulk concentration for the reactants, where the products have zero concentration. The reactant species are reduced to form the products at the electrode boundary. The electrochemical reactions ($NO_3^- + 8e^- + 7H_2O \rightarrow NH_4^+ + 10\,OH^-$ and $H_2O + e^- \rightarrow 1/2\,H_2 + OH^-$) involve the steps in the simulations. The main parameters for the finite element simulations are listed in Supplementary Table S4.

### Data availability

All data generated and analyzed in this study are included in the article and its Supplementary Information and are also available from the authors upon request. Source data are provided with this paper.

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

## Acknowledgements

We acknowledge the National Natural Science Foundation of China (Nos. 22071173 and 22271213 for Y.Y.), the Haihe Laboratory of Sustainable Chemical Transformations for financial support, Tianjin Science and Technology Program (No. 22ZYJDSS00060 for Y.Y.), the Special Fund for Postgraduate Education (B1-2021-010 for Y.H.) and the Guangdong Basic and Applied Basic Research Foundation (No. 2022A1515140051 for Q.L.) for supporting this project. We appreciate Ms Yang Liu in the Analysis and Testing Center at Tianjin University for in situ ATR-FTIR measurements. We acknowledge the National Supercomputing Center in Guangzhou (Sun Yat-Sen University) for calculation source support.

## Author contributions

Y.Y. and Q.L. conceived the idea and directed the research. Y.H., C.H., Q.L., and Y.Y. designed the experiments. Y.H. and C.H. carried out the experiments. C.C. performed the DFT calculations. S.H. and B.Z. provided help in the experiment design. M.H., B.Z., Y.W., and N.M. provided help in the attenuated total reflectance-Fourier transform infrared (ATR-FTIR) spectroscopy and in situ Raman experiments. Y.H., C.H., Q.L., and Y.Y. wrote the paper with comments from all authors.

## Competing interests

The authors declare no competing interests.
