## [Peer review file · Nature Communications]

REVIEWER COMMENTS

Reviewer #1 (Remarks to the Author):

In this manuscript, the authors have performed an electrocatalytic nitrate reduction to ammonia with the pulsed electrocatalysis technique over the RuIn₃/C intermetallic catalysts. With a systematic characterization, the atomic structure of the intermetallic catalyst was determined. The authors introduced pulse potential techniques to solve the dilemmas in low-concentration nitrate reduction. Intriguing catalytic performance was obtained, especially compared with conventional potentiostatic tests. Impressive, The authors elucidate the roles of pulsed potential from the thermodynamics and kinetics perspective using experimental results and theoretical simulations. Considering the deep mechanism study, good electrocatalytic performance, and realistic application potential, the manuscript is well-organized. Therefore, I think this interesting manuscript is suitable for publication in 'Nature Communications' after addressing the following minor concerns.

1. Electrochemical nitrate reduction in an alkaline electrolyte involves eight electrons transfer coupled with the generation of nine OH⁻. So, how about the pH change during the nitrate reduction in alkaline media? It would be better if the pH changes were monitored and discussed.
2. Were the catalysts stable during the nitrate reduction? More characterizations and comparisons are needed for the catalysts before and after the measurements. How about the dissolution of Ru and In during the pulsed nitrate reduction process?
3. As shown in Figs. 5B and C, in situ ATR-FTIR experiments were performed to unveil the role of pulse potential from a thermodynamics perspective. As the ordinate of Fig. 5B is absorption, why is Si-O inverted?
4. As shown in Supplementary Fig. S7, the author provided the LSV curves of RuIn₃ with/without nitrate addition. Supplementing the LSV curves of Ru and In is recommended for better comparison.
5. In Supplementary Fig S5, the author only provides the calibration curve of nitrate. How about its absorbance curves?
6. The quantity of all figs should be improved.
7. Some minor problems. (1) Did the authors use geometric surface area when reporting current densities? (2) The writing quality could be improved, particularly in the methods section. There are places where the meaning could be confused: the authors mix the use of "absorption" and "adsorption", which have different meanings.

Reviewer #2 (Remarks to the Author):

Yu et al. report a novel and effective strategy for electroreduction of low-concentrations NO₃⁻ to ammonia, i.e., pulsed electrolysis technology. The maximal Faradaic efficiency was up to 97%, and the optimal yield rate was about 2.76 mmol/h/mgRu under pulsed potential conditions. Its characterizations, electrochemical measurements, and analysis of reaction mechanisms are persuasive. Overall, this is a high-quality manuscript with a well-organized structure and sufficient discussion. This work will pave a way toward tuning the favorability for other competing electrochemical reactions beyond nitrate reduction to ammonia. I would like to recommend this manuscript to be published in Nature Communications after addressing the following issues.

1. The XRD pattern (Figure 1a) shows that the synthesized sample is RuIn₃ intermetallic compound. In order to gain more insight into the crystal structure and phase purity of the RuIn₃ intermetallic compounds, it is recommended to add related Rietveld refinement results.
2. The surface atomic arrangement of intermetallic compounds has an important effect on the catalytic performance, so the atomic arrangement on the surface of RuIn₃ intermetallic compounds should be analyzed.
3. The catalytic performance is highly associated with the specific surface area of the catalysts. In the manuscript, RuIn₃ is deposited on the carbon as the substrate, but the characterizations about the surface area are missing. The authors should provide these data and corresponding illustrations in the manuscript or supplementary information.

4. The authors mentioned that “the electric energy consumption and production cost for producing 1 kg of NH₃ over RuIn₃/C through pulsed electrolysis were calculated as ~20.2 kW h and US\$0.61 based on the price of renewable electricity (US\$0.03 kWh⁻¹)”. A detailed calculation process should be provided, which is important for the large-scale production of NH₃ by using renewable electricity. Meanwhile, the authors should illustrate the energy cost difference between continuous and pulsed power.

5. The fabricated intermetallic compound seems unique and exhibits excellent catalytic performance compared to reference catalysts. And the authors should illustrate the advantages of the intermetallic compounds as the catalysts from the viewpoints of microstructure and crystal structure.

6. There are some typos in the manuscript, and the format of the references is inconsistent, for example, ref. 7 on page 8, ref. 35 on page 9, refs. 28, 29 and 43 on page 10, ICCD or ICDD indexed in the XRD patterns. The authors should check carefully and correct all the typos throughout the manuscript. Meanwhile, the resolution of some figures is too low, especially for Figures 1, 3b and 4e. Please revise these figures accordingly.

Reviewer #3 (Remarks to the Author):

The manuscript describes the use of alternating pulses to enhance nitrate reduction to ammonia on RuIn₃/C catalysts. Nitrate electrochemical reduction is receiving a lot of attentions from the research community as a green alternative for the production of ammonia. Despite the interest of the reaction reported, there are fundamental aspects that are not properly addressed making the quality of manuscript below what is expected for Nature Communications.

First, one of the major drawbacks for the synthesis of ammonia from nitrates in polluted waters is its low concentration. The authors claim that the main advantage of such method is its applicability in low nitrate concentrations, however, if low concentrations of nitrate are aimed (for water purification), a product other than ammonia should be aimed.

Another important aspect, is the novelty. Chronoamperometric pulses have already been explored for nitrate reduction to ammonia in a previous publication (Journal of the American Chemical Society 2023 145 (11), 6471-6479). Although the catalysts are different the approach is similar, reducing significantly the novelty of the described work.

A collection of (some) other issues to address is given below:

- Pulsed electrocatalysis is not a technique. Electrocatalysis is a branch of electrochemistry and the technique used chronoamperometric pulses.
- The authors mention that one of the advantages of this study is to help on the rational design and manipulation of catalysts, however according with the results there are no changes in the catalysts.
- The Tafel slope calculated is meaningless. For a proper Tafel analysis, several assumptions are necessary. The most important one is that the reaction can not be mass transfer limited, so the analysis is confined to low currents and low overpotentials. This is not the case in the manuscript and such a high Tafel slope confirms the mass transfer limitations and not the limited adsorption and activation of nitrate.
- For the optimization of the pulse parameters, the motivation for the pulse length studied should be given. In some situations, the results seem to not be statistically different.
- DEMS experiments. How were the experiments performed? It is strange that the intensities follow similar trends with and without pulses. It would be expected that for the static conditions the signals from the products detected would increase or be constant over time and not show pulses.
- ATR-FTIR. For signal enhancement, the window was covered with the Au layer. Au is known to enhance nitrate reduction (RSC Adv. 2023 Mar 27; 13(15): 9839–9844.). The authors should also perform an experiment without the RuIn₃ catalyst, in the presence of nitrate.
- It is not surprising that by applying a positive pulse the negatively charged species increase their concentration in the vicinity of the electrode. This can definitely help on replenishing the electrode surface with reactive species for further reduction when the potential is negative again. The fundamental aspect that increases the FE is the enhanced mass transport. The authors should

compare the results obtained with the chronoamperometric pulses with other mass transport enhancement phenomena such as RDE or electrolysis under flow conditions.

A point-by-point response to the reviewers' comments

To Reviewer 1:

Reviewer Letter: In this manuscript, the authors have performed an electrocatalytic nitrate reduction to ammonia with the pulsed electrocatalysis technique over the RuIn₃/C intermetallic catalysts. With a systematic characterization, the atomic structure of the intermetallic catalyst was determined. The authors introduced pulse potential techniques to solve the dilemmas in low-concentration nitrate reduction. Intriguing catalytic performance was obtained, especially compared with conventional potentiostatic tests. Impressive, the authors elucidate the roles of pulsed potential from the thermodynamics and kinetics perspective using experimental results and theoretical simulations. Considering the deep mechanism study, good electrocatalytic performance, and realistic application potential, the manuscript is well-organized. Therefore, I think this interesting manuscript is suitable for publication in 'Nature Communications' after addressing the following minor concerns.

Answer: We highly appreciate the reviewer for the positive comments on our work. To save the reviewer's valuable time, key revisions are displayed in a yellow background in the revised manuscript and Supplementary Information.

Comment 1: Electrochemical nitrate reduction in an alkaline electrolyte involves eight electrons transfer coupled with the generation of nine OH⁻. So, how about the pH change during the nitrate reduction in alkaline media? It would be better if the pH changes were monitored and discussed.

Answer 1: The pH value of the electrolyte during pulsed electrocatalytic nitrate reduction to ammonia (NRA) process ($E_a = +0.6$ V, $t_a = 0.5$ s, $E_c = -0.1$ V, $t_c = 4$ s) was monitored (**Figure R1b, revised supplementary Fig.12b**). The pH value changed a little from 12.78 to 12.93 due to the low concentration of nitrate, indicating the stability of the reaction system. Combined with the cycle stability tests, XRD patterns, TEM images and the corresponding particle size distribution, the stability of reaction system and the catalyst in the pulsed NRA process can be proved (**Figure R1, revised supplementary Fig.12**).

Figure R1 (revised supplementary Fig. 12) (a) Cyclic durability test of RuIn₃/C for 10 mM nitrate electroreduction to ammonia under pulsed conditions ($E_c = -0.1 \text{ V}$, $E_a = +0.6 \text{ V}$, $t_c = 4 \text{ s}$, $t_a = 0.5 \text{ s}$). (b) The change of pH value of electrolyte during the pulsed NRA process ($E_c = -0.1 \text{ V}$, $E_a = +0.6 \text{ V}$, $t_c = 4 \text{ s}$, $t_a = 0.5 \text{ s}$). (c) The XRD patterns of RuIn₃/C before and after the cycle stability test under pulsed conditions. (d) TEM image of the RuIn₃/C catalysts after the stability test. (e) The corresponding particle size distribution.

Comment 2: Were the catalysts stable during the nitrate reduction? More characterizations and comparisons are needed for the catalysts before and after the measurements. How about the dissolution of Ru and In during the pulsed nitrate reduction process?

Answer 2: First, the performance stability of RuIn₃/C catalyst for pulsed NRA was examined at $E_a = +0.6$ V, $t_a = 0.5$ s, $E_c = -0.1$ V, $t_c = 4$ s in 0.1 M KOH solution containing 10 mM nitrate for 10 cycles. The ammonia Faradaic efficiency and yield rate remained well throughout the ten-cycle stability tests (**Figure R1a, revised supplementary Fig.12a**). Then, the pH value of the electrolyte changed a little from 12.78 to 12.93 (**Figure R1b, revised supplementary Fig.12b**). XRD patterns, TEM image, and the corresponding particle size distribution (**Figure R1c-e, revised supplementary Fig.12c-e**) of the utilized RuIn₃/C samples remained unchanged. These results demonstrated the excellent stability of RuIn₃/C catalysts. In addition, the ion concentrations of Ru and In in the electrolyte were quantified by the ICP-OES as well, and no Ru or In ions were detected.

Comment 3: As shown in Figs. 5B and C, *in situ* ATR-FTIR experiments were performed to unveil the role of pulse potential from a thermodynamics perspective. As the ordinate of Fig. 5B is absorbance, why is Si-O inverted?

Answer 3: The *in situ* ATR-FTIR spectra were acquired in absorbance mode, in which the upward peaks represented the generated intermediates. The increase of upward peaks indicated the accumulation of the generated intermediates. As illustrated in **Figure R2**, the Si-O bonds were downward peaks because the amount of Si-O bonds was continuously decreased throughout the electrocatalytic reduction process.

Figure R2. Time-dependent *in situ* ATR-FTIR spectra of NRA under different conditions. Potentiostatic ($E = -0.1$ V) using (a) 0.1 M KOH containing 10 mM nitrate (**Fig. 5b in the revised manuscript**) and (b) 0.1 M KOH (**revised supplementary Fig. 30b**). Pulsed potential ($E_c = -0.1$ V, $E_a = +0.6$ V, $t_c = 4$ s, $t_a = 0.5$ s) using (c) 0.1 M KOH containing 10 mM nitrate (**Fig. 5c in the revised manuscript**) and (d) 0.1 M KOH as electrolyte (**revised supplementary Fig. 30a**).

Comment 4: As shown in Supplementary Fig. S7, the author provided the LSV curves of RuIn₃ with/without nitrate addition. Supplementing the LSV curves of Ru and In is recommended for better comparison.

Answer 4: According to the reviewer's suggestion, the LSV curves of Ru/C and In/C with and without nitrate addition were added (**Figure R3, revised supplementary Fig. 6**). In/C showed neglectable current enhancement after adding 10 mM nitrate, demonstrating its poor NRA activity. RuIn₃/C and Ru/C exhibited a noticeable increase in current density after adding 10 mM nitrate, indicating the electroreduction of nitrate. Notably, after normalization to the electrochemical active surface area (ECSA), RuIn₃/C still showed a higher current density than Ru/C, proving its high intrinsic activity (**Figure R4, revised supplementary Fig. 17**).

Figure R3 (revised supplementary Fig. 6). LSV curves of various catalysts in 0.1 M KOH solution with and without 10 mM nitrate under 1800 rpm with 85% iR correction.

Figure R4 (revised Supplementary Fig. 17). CV curves of (a) In/C, (b) Ru/C, (c) RuIn₃/C with various scan rates from 5 mV s⁻¹ to 100 mV s⁻¹. (d) Fitting lines of the current density versus the different scan rates, (e) the calculated ECSA, and (f) the LSV curves normalized to the ECSA for distinct catalysts.

Comment 5: In Supplementary Fig S5, the author only provides the calibration curve of nitrate. How about its absorbance curves?

Answer 5: According to the reviewer's kind suggestion, the absorbance curves of nitrate, nitrite, and ammonia were provided in **Figure R5 (revised supplementary Fig. 5)** and added in the revised Supplementary information. The calibration curves of NO₃⁻, NO₂⁻ and NH₄⁺ showed good linear relationship.

Figure R5 (revised supplementary Fig. 5). (a) UV-vis absorption spectra and (b) the corresponding calibration curve for NO_3^- . (c) UV-vis adsorption spectra and (d) the corresponding calibration curve for NO_2^- . (e) UV-vis absorption spectra and (d) the corresponding calibration curve for NH_4^+ .

Comment 6: The quantity of all figs should be improved.

Answer 6: All the figures in the revised manuscript and supplementary information have been reprocessed. We believe that the quality of these figures can meet the high requirements of the journal.

Institute of Molecular Plus
Tianjin University
Tianjin 300072, P. R. China
Tel & Fax: 86-22-27403475
E-mail: yyu@tju.edu.cn

Comment 7: Some minor problems. (1) Did the authors use geometric surface area when reporting current densities? (2) The writing quality could be improved, particularly in the methods section. There are places where the meaning could be confused: the authors mix the use of "absorption" and "adsorption", which have different meanings.

Answer 7: We thank the reviewer for these kind comments and suggestions. The current density reported in this work is based on the geometric surface area. We have clarified this point in the **Electrochemical Measurements parts** in the revised manuscript. Meanwhile, we have carefully rewritten the **Methods** section to enhance its clarity and readability. Sorry for the mistaken use of adsorption and absorption. We have checked the manuscript and corrected these mistakes.

We acknowledge again for all the kind comments and wise suggestions from you. We are sure that the quality of this work will be greatly improved according to these helpful comments and wise suggestions.

To Reviewer 2

Reviewer Letter: Yu et al. report a novel and effective strategy for the electroreduction of low-concentration NO_3^- to ammonia, i.e., pulsed electrolysis technology. The maximal Faradaic efficiency was up to 97%, and the optimal yield rate was about 2.76 mmol/h/mgRu under pulsed potential conditions. Its characterizations, electrochemical measurements, and analysis of reaction mechanisms are persuasive. Overall, this is a high-quality manuscript with a well-organized structure and sufficient discussion. This work will pave a way toward tuning the favorability for other competing electrochemical reactions beyond nitrate reduction to ammonia. I would like to recommend that this manuscript to be published in Nature Communications after addressing the following issues.

Answer: We highly appreciate the reviewer for the positive comments on our work. To save the reviewer's valuable time, key revisions are displayed in a yellow background in the revised manuscript and Supplementary Information.

Comment 1: The XRD pattern (Figure 1a) shows that the synthesized sample is RuIn_3 intermetallic compound. In order to gain more insight into the crystal structure and phase purity of the RuIn_3 intermetallic compounds, it is recommended to add related Rietveld refinement results.

Answer: According to the reviewer's kind suggestion, we have added the related Rietveld refinement results in the revised manuscript (**Fig.1a in the revised manuscript**). The corresponding Rietveld refinement result revealed the high phase purity of the obtained material (**Figure R6, Fig.1a in the revised manuscript**). Furthermore, the refined lattice parameters were summarised in **Table R1 (revised Supplementary Table 1)**."

Figure R6 (Fig.1a in the revised manuscript). The XRD pattern and Rietveld refinement of RuIn_3/C .

Table R1 (revised Supplementary Table S1). Unit cell parameters of RuIn_3 calculated from the XRD data.

RuIn ₃	
Space group	P42/mnm
a	6.997 Å
b	6.997 Å
c	7.242 Å
α	90°
β	90°
γ	90°

Comment 2: The surface atomic arrangement of intermetallic compounds has an important effect on the catalytic performance, so the atomic arrangement on the surface of RuIn₃ intermetallic compounds should be analyzed.

Answer: During the thermal annealing process, the RuIn₃ intermetallic compounds tended to expose the facets with low surface energies and became spheres to minimize the total surface energy. Based on the calculation results of surface energy (**Figure R7a**), the (112), (202) and (212) facets of RuIn₃ exhibit relatively low surface energies of 1.02, 1.03, and 0.98 J m⁻², respectively, making them to be the predominant exposed facets. Meanwhile, in **Figure R7b**, AC-HAADF-STEM reveals the observed lattice spacings of 2.51, 3.12 and 2.36 Å, which can be indexed to the (202), (210) and (212) facets, respectively. These two results are consistent with each other. Furthermore, the atomic arrangements of these three facets are listed in **Figure R8**.

Figure R7. (a) DFT calculations on the surface energy of different RuIn₃ facets and (b) AC-HAADF-STEM image of RuIn₃/C.

Figure R8. The geometric structures in the top view of (a) (202), (b) (210) and (c) (212) crystal facets of RuIn₃.

Comment 3: The catalytic performance is highly associated with the specific surface area of the catalysts. In the manuscript, RuIn₃ is deposited on the carbon as the substrate, but the characterizations about the surface area are missing. The authors should provide these data and corresponding illustrations in the manuscript or supplementary information.

Answer: We acknowledge the valuable comments. The electrochemical active surface area (ECSA) was added to ascertain the specific active surface areas for comparing the activities of distinct catalysts. The ECSA was assessed through the double-layer capacitance method in 0.1M KOH solution within the non-Faradaic potential range, employing various scan rates ranging from 5 to 100 mV s⁻¹. The ECSA of the working electrodes was calculated according to the following equations:

$$I_c = \nu C_{dl}$$

$$ECSA = \frac{C_{dl}}{C_s}$$

Where the I_c represents the charging current with different scan rates, ν is the scan rate, and C_{dl} is the double-layer capacitance. C_s represents the specific capacitance for a flat metallic surface, generally during the 20-60 $\mu\text{F cm}^{-2}$ range. According to the reported literature, we assumed C_s was 40 $\mu\text{F cm}^{-2}$ (*J. Am. Chem. Soc.*, 2013, 135, 16977–16987; *Adv. Energy Mater.*, 2019, 9, 1900149; *J. Am. Chem. Soc.*, 2015, 137, 4347–4357). Subsequently, the electric double-layer capacitance test showed that the ECSA of Ru/C, In/C and RuIn₃/C was 56.75, 116 and 140.75 cm_{ECSA}^2 , respectively (**Figures R9a-e, revised Supplementary Figs. 17a-e**). Notably, after the LSV curves of the electrocatalysts were normalized to the ECSA, RuIn₃/C possessed the highest current density (**Figure R9f, revised Supplementary Fig. 17f**), indicating its highest intrinsic activity.

Figure R9 (Supplementary Fig. 17). CV curves of (a) In/C, (b) Ru/C, (c) RuIn₃/C with various scan rates from 5 mV s⁻¹ to 100 mV s⁻¹. (d) Fitting lines of the current density versus the different scan rates, (e) the calculated ECSA, (f) the LSV curves normalized to the ECSA for distinct catalysts.

Comment 4: The authors mentioned that "the electric energy consumption and production cost for producing 1 kg of NH₃ over RuIn₃/C through pulsed electrolysis were calculated as ~20.7 kW h and US\$0.62 based on the price of renewable electricity (US\$0.03 kWh⁻¹)". A detailed calculation process should be provided, which is important for the large-scale production of NH₃ by using renewable electricity. Meanwhile, the authors should illustrate the energy cost difference between continuous and pulsed power.

Answer: The calculation details of electric energy consumption and production cost for 1 kg NH₃ were referenced to the reported literature (*J. Am. Chem. Soc.* 2020, 142, 5702-5708; *ACS Catal.* 2023, 13, 6268-6279) and added in the revised Supplementary information.

The electricity cost for producing 1 kg NH₃ per day from pulsed NRA was calculated as follows.

The partial current (I_{NH_3}) for production 1 kg NH₃ was calculated as:

$$I_{NH_3} = \frac{Q_{NH_3}}{t} = \frac{zFn_{NH_3}}{t}$$

Where z is the number of electrons transferred in nitrate reduction to ammonia. F is the Faraday constant. t is the time.

The total current needed is then given by dividing the Faradaic efficiency:

$$I_{total} = \frac{I_{NH_3}}{FE}$$

The required power is given from $P=UI$:

$$power = P = U \times I = cell\ potential \times I_{total}$$

$$U = cell\ potential = E_{OER} - E_{NRA}$$

We assumed that the overpotential of the water oxidation is zero and there are no ohmic losses. Thus, E_{OER} is 1.23 V. E_{NRA} is the experimental value of the cathodic potential. Thus, the cell potential ($E_{OER} - E_{NRA}$) equals (1.23- E_{NRA}) V.

Taking into account the distinctive characteristics of the pulse process and drawing insights from the studies of Gu et al. (*J. Am. Chem. Soc.* 2023, 145, 2195-2206), the effective cathodic energy consumption efficiency ($E_{cathodic_consum}$) was calculated as follows:

$$E_{cathodic_consum} = \frac{|Q_{cathodic}|}{|Q_{cathodic}| + |Q_{anodic}|}$$

The effective cathodic energy efficiency at $E_a = +0.6$ V, $t_a = 0.5$ s, $E_c = -0.1$ V, and $t_c = 4$ s was calculated to be 82.8%. Some additional energy (about 17.2%) was used during the auxiliary oxidation or prereduction step.

The electricity power requirement is calculated from the power and the total time (1 day) after considering the energy waste:

$$\text{Electricity requirement} = \frac{\text{power} \times 24 \text{ hour}}{E_{\text{cathodic_consum}}}$$

The electricity cost is calculated from the electricity requirement and the price of electricity:

$$\text{Electricity cost} = \text{Electricity requirement} \times \text{price}$$

The cost of electricity was calculated based on the price of renewable electricity alone (US\$ 0.03 kWh⁻¹) (*Nat. Catal.* 2020, 3, 125-134), the energy consumption and production cost per one kilogram of NH₃ over RuIn₃/C through pulsed electrolysis were calculated as ~20.7 kWh and US\$ 0.62. Note that this is a simple cost accounting based on electricity price, without considering capital costs and Ohmic losses. Considering the environmental benefit of nitrate contaminant removal, the reported RuIn₃/C for pulsed NRA is very appealing.

Comment 5: The fabricated intermetallic compound seems unique and exhibits excellent catalytic performance compared to reference catalysts. And the authors should illustrate the advantages of the intermetallic compounds as the catalysts from the viewpoints of microstructure and crystal structure.

Answer: Thanks for your valuable suggestion. In the RuIn₃ intermetallic compounds with tetragonal crystal structure, Ru is bonded in an 8-coordinate geometry to eight In atoms. The excellent electrocatalytic properties of the fabricated RuIn₃ intermetallic compounds can be rationally attributed to the distinctive crystal structure. Alloying Ru with the relatively low electronegative In to form an ordered intermetallic phase can modulate the electronic structure of Ru sites and enhance the catalytic performance (*Handbook of Chemistry and Physics*, 85th Edition, CRC Press, 2004; *Adv. Energy Mater.* 2022, 12, 2270108; *ACS Catal.*, 2022, 12, 2623; *Angew. Chem. Int. Ed.*, 2022, 61, e202202017).

Comment 6: There are some typos in the manuscript, and the format of the references is inconsistent, for example, ref. 7 on page 8, ref. 35 on page 9, refs. 28, 29 and 43 on page 10, ICCD or ICDD indexed in the XRD patterns. The authors should check carefully and correct all the typos throughout the manuscript. Meanwhile, the resolution of some figures is too low, especially for Figures 1, 3b and 4e. Please revise these figures accordingly.

Answer: We thank the reviewer for these comments and suggestions. We have carefully checked and corrected the mistakes in the revised manuscript and supplementary information.

Institute of Molecular Plus
Tianjin University
Tianjin 300072, P. R. China
Tel & Fax: 86-22-27403475
E-mail: yyu@tju.edu.cn

We acknowledge again for all the kind comments and wise suggestions from you. We are sure that the quality of this work will be greatly improved according to these helpful comments and wise suggestions.

To Reviewer 3:

Reviewer Letter: The manuscript describes the use of alternating pulses to enhance nitrate reduction to ammonia on RuIn₃/C catalysts. Nitrate electrochemical reduction is receiving a lot of attentions from the research community as a green alternative for the production of ammonia. Despite the interest of the reaction reported, there are fundamental aspects that are not properly addressed making the quality of manuscript below what is expected for Nature Communications.

First, one of the major drawbacks for the synthesis of ammonia from nitrates in polluted waters is its low concentration. The authors claim that the main advantage of such method is its applicability in low nitrate concentrations, however, if low concentrations of nitrate are aimed (for water purification), a product other than ammonia should be aimed.

Another important aspect, is the novelty. Chronoamperometric pulses have already been explored for nitrate reduction to ammonia in a previous publication (*Journal of the American Chemical Society* 2023 145 (11), 6471-6479). Although the catalysts are different the approach is similar, reducing significantly the novelty of the described work.

Answer: Thanks for the reviewer's comments on our manuscript. As for the reviewer's concerns, we have provided a point-by-point response. To save the reviewer's valuable time, key revisions are displayed on a yellow background in the revised manuscript and Supplementary Information.

Reviewer 3 did not suggest the acceptance of our work because of the following two major questions: (1) One of the major drawbacks for the synthesis of ammonia from nitrates in polluted waters is its low concentration; (2) chronoamperometric pulses have already been explored for nitrate reduction to ammonia in a previous publication. We would like to explain them one by one:

(1) The electrochemical reduction of nitrate into ammonia has two promising application areas. One is the reduction of high-concentration nitrate to high-concentration ammonia, which will be taken out as NH₃ product (*Nat. Nanotechnol.* 2022, 17, 759–767). The other is the reduction of low-concentration nitrate to **low-concentration ammonia** solution, which serves as **ammonia-containing liquid fertilizer**. Recently, we published a **perspective** entitled “Sustainable production and in-place utilization of a liquid nitrogenous fertilizer” (*Joule*, 2023, DOI: 10.1016/j.joule.2023.07.020) to elaborate on its application potential. As known, in conventional agriculture, solid nitrogenous fertilizer is manufactured in centralized chemical plants and transported to where needed (*Joule* 2018, 2, 1055-1074.). These procedures are highly dependent on fossil fuels, leading to the emission of vast amounts of greenhouse gases. Moreover, the utilization rate of the applied solid nitrogenous fertilizer is less than 50%, leading to the leaching of unabsorbed nitrogenous fertilizer into underground water and causing inevitable resource waste (*Ecol. Eng.* 2017, 107, 8-32). In our perspective, we proposed a tandem reaction to prepare liquid nitrogenous fertilizer from the air. In the former reaction, the air was driven by plasma/photocatalysis to produce NO_x, which was absorbed by water to generate **low-concentration nitrate**. In the latter reaction, the

electrochemical technology converted low-concentration nitrate to **low-concentration ammonia solution**, which served as ammonia-containing liquid fertilizer. This tandem reaction can be operated under variable power sources. Therefore, this technique can match well with renewable solar and wind energy. **In this regard, the electroreduction of low-concentration nitrate to low-concentration ammonia is highly important.**

(2) **The crucial goal of catalysis research is to develop efficient catalysis systems to promote and even alter the rate-limiting step based on the understanding of the reaction mechanism.** As the reviewer claimed, Li et al. reported an interesting tandem catalysis process for NRA, in which **two alternated negative potentials** were used (*J. Am. Chem. Soc.* 2023, 145, 6471-6479.). Under the first negative potential, nitrate was reduced to nitrite. Under the second negative potential, nitrite was reduced to ammonia. Both negative potentials were used to reduce substance. **The novelty of our work lies in the rational design and manipulation of reaction processes for low-concentration nitrate reduction. For the first time**, a pulsed positive voltage was adopted to replenish the negatively charged NO_3^- in the vicinity of the working electrode and suppress the competitive hydrogen evolution reaction. A NRA performance (Faradaic efficiency: 97.6%, yield rate: $2.7 \text{ mmol}^{-1} \text{ h}^{-1} \text{ mg}_{\text{Ru}}^{-1}$, conversion rate: 96.4%) was achieved for low-concentration ($\leq 10 \text{ mM}$) nitrate reduction under **the periodic alternated positive and negative potentials**. Interestingly, the nitrate concentration can be continuously reduced to 0.35 mM under pulsed conditions, lower than the emission standard ($\leq 0.81 \text{ mM}$). Furthermore, the combined results of mass-transfer analysis (*in situ* Raman spectroscopy and finite element analysis) and reaction pathway analysis (online differential electrochemical mass spectroscopy and *in situ* ATR-FTIR) reveal that the **periodic alternated** positive and negative potentials can promote the accumulation of nitrate near the electrode and **favor the formation of a more active *NO intermediate (on-top absorption configuration)**, which synergistically promote the reduction of low-concentration nitrate to ammonia. **Thus, the enhancement mechanism of the proposed pulsed positive and negative potentials is novel and can be extended to other electrochemical reactions.**

We believe that our manuscript is a meaningful research with unusual urgency and significance in electrocatalysis that appeals to a broad, general audience in the multidisciplinary fields of chemistry, materials, environmental, and energy. Thus, **Reviewers 1 and 2 gave very positive comments on the novelty and importance of our work** and provided professional revision suggestions. **Reviewer 1** said, "Considering the **deep mechanism study, good electrocatalytic performance, and realistic application potential**, the manuscript is **well-organized**. Therefore, I think this interesting manuscript is suitable for publication in '**Nature Communications**' after addressing the following minor concerns." **Reviewer 2** mentioned, "Overall, this is a **high-quality** manuscript with a **well-organized structure and sufficient discussion**. This work will pave a way toward tuning the favorability for other competing electrochemical reactions beyond nitrate reduction to ammonia".

Moreover, in response to the third reviewer's kind concerns, we have supplemented the rotating disk electrode tests, *in situ* ATR FTIR, and online DEMS experiments in the revised manuscript and supporting information. The details of related experiments have been added to the manuscript and Supplementary Information to facilitate reference and repetition. Thus, our revised manuscript is a greatly improved version, which will meet the high criteria of "**Nature Communications**".

Comment 1: Pulsed electrocatalysis is not a technique. Electrocatalysis is a branch of electrochemistry, and the technique uses chronoamperometric pulses.

Answer 1: According to the reviewer's kind suggestion, we have replaced the relevant expression of "pulsed technique" with "pulsed approach" in the revised manuscript and supporting information.

Comment 2: The authors mention that one of the advantages of this study is to help with the rational design and manipulation of catalysts. However, according to the results, there are no changes in the catalysts.

Answer 2: Thanks for this important comment here. *In situ* ATR-FTIR experiments were performed to reveal the function of pulsed potential from the thermodynamic perspective. The extra bands at $\sim 1690\text{ cm}^{-1}$ and $\sim 1740\text{ cm}^{-1}$ (designated to on-top adsorption of NO (*J. Catal.* 2007, 249, 311-317; *Langmuir* 2008, 24, 4352-4357)) can only be observed under pulsed conditions (**Figures R10a and b, Figs.5b and c in the revised manuscript**). Based on the information offered by online DEMS, *in situ* ATR-FTIR, and reported literature, the possible reaction pathways were deduced as N-end pathway ($*\text{NO}_3 \rightarrow *\text{NO}_2 \rightarrow *\text{NO} \rightarrow *\text{NOH} \rightarrow *\text{N} \rightarrow *\text{HN} \rightarrow *\text{H}_2\text{N} \rightarrow *\text{NH}_3$) (**Figure R10c, Fig.5e in the revised manuscript**). NO has been considered as the critical intermediate in NRA. It has been reported that the reduction rate of on-top adsorbed NO is higher than that of bridged adsorbed NO (*J. Phys. Chem. C* 2010, 114, 6011-6018; *Langmuir* 2008, 24, 4352-4357; *Chem. Rev.* 2009, 109, 2209-2244.). Thus, pulsed electrolysis can promote the formation of a more active NO absorption configuration, enhancing NRA performance. To avoid misunderstanding, the related expression was modified: "Furthermore, our research implies a novel approach for the rational design and precise manipulation of **reaction processes**, potentially extending its applicability to a broader range of electrocatalytic reactions."

Figure R10 (Figs. 5b, c, and e in the revised manuscript). Time-dependent *in situ* ATR-FTIR spectra of NRA under (a) constant potential ($E = -0.1$ V), (b) pulsed potential ($E_c = -0.1$ V, $E_a = +0.6$ V, $t_c = 4$ s, $t_a = 0.5$ s). (c) Schematic illustration for the NRA reaction pathways over RuIn₃/C.

Comment 3: The Tafel slope calculated is meaningless. For a proper Tafel analysis, several assumptions are necessary. The most important one is that the reaction can not be mass transfer limited, so the analysis is confined to low currents and low overpotentials. This is not the case in the manuscript and such a high Tafel slope confirms the mass transfer limitations and not the limited adsorption and activation of nitrate.

Answer 3: According to the reviewer's kind suggestion, the related Tafel slope analysis was removed in the revised manuscript. Moreover, the LSV curves of Ru/C and In/C with and without nitrate addition were added (**Figure R11, revised supplementary Fig. 6**). In/C showed neglectable current enhancement with nitrate, demonstrating its poor NRA activity. RuIn₃/C and Ru/C exhibited a noticeable increase in current density after adding 10 mM nitrate, indicating the presence of nitrate reduction. Notably, after normalization to the electrochemical active surface area (ECSA), RuIn₃/C still showed a higher current density than Ru/C, proving its high intrinsic activity (**Figure R12, revised supplementary Fig. 17**).

Figure R11 (revised supplementary Fig. 6). LSV curves of various catalysts in 0.1 M KOH solution with and without 10 mM nitrate under 1800 rpm with 85% iR correction.

Figure R12 (revised Supplementary Fig. 17). CV curves of (a) In/C, (b) Ru/C, (c) RuIn₃/C with various scan rates from 5 mV s⁻¹ to 100 mV s⁻¹. (d) Fitting lines of the current density versus the different scan rates, (e) the calculated ECSA, and (f) the LSV curves normalized to the ECSA for distinct catalysts.

Comment 4: For the optimization of the pulse parameters, the motivation for the pulse length studied should be given. In some situations, the results seem to not be statistically different.

Answer 4: We highly appreciate the reviewer's comment. It is essential to optimize the applied potential and its duration time during the pulsed NRA process for pursuing higher ammonia Faradaic efficiency and yield rate. As the applied potential went more negative, the ammonia Faradaic efficiency over RuIn₃/C increased first and then decreased, while the ammonia yield rate continuously increased (**Figure R13a, revised Supplementary Fig. 9a**). Hence, -0.1 V was selected for the following experiments. Considering the sluggish kinetics of nitrate reduction to ammonia, the cathodic durations were set to be 2s, 4s, and 8s. When the cathodic and anodic times were constant to screen the optimal anodic potential, $E_a=+0.6$ V displayed the highest ammonia Faradaic efficiency and yield rate (**Figures R13b-d, revised Supplementary Figs. 9b-d**). Thus, +0.6 V was chosen as the optimal anodic potential. Likewise, $t_a=0.5$ s and $t_c=4$ s were chosen. As the reviewer claimed, the results seem not to be totally different. The applied potential and duration time were chosen based on comprehensive NRA performance. Ultimately, the parameters of $E_c=-0.1$ V, $E_a=+0.6$ V, $t_c=4$ s, and $t_a=0.5$ s were screened for the following experiment.

Figure R13 (revised Supplementary Fig. 9). Screening of optimal pulse parameters. (a) Potentiostatic conditions with different applied potentials. Screening of different anodic and cathodic times at (b) $E_a = 0.4$ V, (c) $E_a = 0.6$ V, (d) $E_a = 0.8$ V.

Comment 5: DEMS experiments. How were the experiments performed? It is strange that the intensities follow similar trends with and without pulses. It would be expected that for the static conditions the signals from the products detected would increase or be constant over time and not show pulses.

Answer 5: Thanks for this important comment here. The schematic illustration of the DEMS experiment is provided in **Figure R14a (revised Supplementary Fig. 28a)**, and the more detailed experiment process was added in the revised supporting information. The online DEMS measurements were performed on a QAS 100 (Linglu Instruments (Shanghai) Co. Ltd) to capture the volatile intermediates and products. For the online DEMS test, carbon papers coated with catalysts, platinum wire, and Hg/HgO (filled with fresh 0.1 M KOH) were employed as the working, counter, and reference electrodes, respectively. 0.1 M KOH containing 0.01 M NO_3^- kept flowing through the homemade electrochemical cell through a peristaltic pump. The signal was gathered using a hydrophobic polytetrafluoroethylene (PTFE) membrane, essential for allowing volatile compounds while preventing water entry into the vacuum chamber. The produced

volatile products were brought to the mass spectrometer through a pump. The DEMS tests included two modes. The first mode was the continuous power-on mode with continuous constant potential ($E = -0.1$ V) or continuous pulsed potential ($E_c = -0.1$ V, $E_a = +0.6$ V, $t_c = 4$ s, $t_a = 0.5$ s) (**Figures R14b and c, revised Supplementary Figs. 28b and c**). The second mode was the power switch on and off mode (**Figure R15**). The second mode included three stages. In the first stage, the switch-on potential was constant ($E = -0.1$ V), and the process of switching on and off was repeated 6 times. In the second stage, the switch-on potential was pulsed ($E_c = -0.1$ V, $E_a = +0.6$ V, $t_c = 4$ s, $t_a = 0.5$ s), and the process of switching on and off was repeated 6 times. In the third stage, the switch-on potential is constant ($E = -0.1$ V), and the process of switching on and off was repeated 6 times. The duration time for switch on and off are 50s and 180s, respectively. Notably, the duration time for DEMS is much longer than that of a pulse potential ($t_c = 4$ s, $t_a = 0.5$ s). Thus, detecting product signals in one switch contained several pulse cycles. Adopting the alternative switch on and off mode for the DEMS test was a common method, which has been widely reported (*Nat. Catal.* 2021, 5, 66-73; *Nat. Catal.* 2023, 6, 402-414).

Figure R14 (revised Supplementary Fig. 28). (a) Schematic illustration for the online DEMS electrochemical measurement. The enlarged signal intensity of $m/z=2$ under the application of continuous (b) constant potential ($E = -0.1$ V) and (c) pulsed potential ($E_c = -0.1$ V, $E_a = +0.6$ V, $t_c = 4$ s, $t_a = 0.5$ s).

Figure R15. Online DEMS signal of $m/z=2$ during the NRA process, starting with a constant potential ($E = -0.1$ V), then switching to a pulsed potential ($E_c = -0.1$ V, $E_a = +0.6$ V, $t_c = 4$ s, $t_a = 0.5$ s), and finally switching back to the starting constant potential.

Comment 6: ATR-FTIR. For signal enhancement, the window was covered with the Au layer. Au enhances nitrate reduction (RSC Adv. 2023 Mar 27; 13(15): 9839–9844.). The authors should also perform an experiment without the RuIn3 catalyst in the presence of nitrate.

Answer 6: Thank the reviewer for the comments. To address the reviewer's concern, the LSV curves of the Au plate in 0.1 M KOH solution with and without 10 mM nitrate were recorded in an H-cell (**Figure R16a**). With the addition of 10 mM nitrate, the current densities changed slightly. Moreover, the entire current density was limited at several milliampere levels, indicating the poor activity of nitrate reduction over the Au surface, consistent with the reported literature (ACS Catal. 2019, 9, 7052-7064; J. Electroanal. Chem. 1996, 414, 163-170; Phys. Chem. Chem. Phys., 2013, 15, 3196-3202). The mentioned Au-doped Cu catalyst (RSC Adv. 2023; 13, 9839–9844.) by the reviewer only showed obvious nitrate reduction to ammonia performance negative than -0.7V. The doped Au facilitated electron transfer, and Au was not explicitly identified as the active center. In addition, Cu has been regarded as a highly active nitrate reduction catalyst (J. Am. Chem. Soc. 2020, 142, 5702-5708; Nanoscale 2020, 12, 9385-9391; Angew. Chem. Int. Ed. 2020, 59, 5350-5354). It would be more proper to regard Au as a promotion in the mentioned literature. Then, *in situ* ATR FTIR experiments were carried out on an Au-coated Si prism in a 0.1 M KOH + 10 mM nitrate solution ranging from +0.3 V to -1.2 V vs. RHE (**Figure R16b**). Only two absorption bands of water molecule bending (~ 1630 cm^{-1}) and Si-O stretching (~ 1238 cm^{-1}) were observed between +0.3 and -0.55 V vs. RHE. The bands of absorbed N-H at ~ 1427 cm^{-1} were very low, and the intensity increased as the potential went more negative. However, the bands of N-containing species

assigned to N-H, N-O-H, and NO were detected over RuIn₃/C covered Au coated Si prism above +0.2 V vs. RHE (**Figure R16c**). For better comparison, the *in situ* ATR-FTIR spectra of Au coated Si prism and RuIn₃/C covered Au coated Si prism were selected and compared (**Figure R16d**). At the same potential, especially at the low overpotential part, the signal intensities of N-related intermediates obtained over RuIn₃/C were higher than that obtained over Au coated Si prism. These findings suggest that the nitrate electroreduction proceeded on the RuIn₃/C catalyst rather than Au, and N-related bonds originate from nitrate reduction rather than contaminants.

Figure R16. (a) j - V plots of Au plate and RuIn₃/C in 0.1 M KOH solution with and without 10 mM nitrate (85% iR correction). Potential-dependent *in situ* ATR-FTIR spectra of NRA over (b) Au coated Si prism, and (c) RuIn₃/C @ Au coated Si prism using 0.1 M KOH containing 10 mM nitrate as electrolyte. (d) The selected *in situ* ATR-FTIR spectra obtained at +0.3, 0, -0.3, and -1.2 V from Au coated Si prism and RuIn₃/C @ Au coated Si prism.

Comment 7: It is not surprising that by applying a positive pulse the negatively charged species increase their concentration in the vicinity of the electrode. This can definitely help replenish the electrode surface with reactive species for further reduction when the potential is negative again. The fundamental aspect that increases the FE is enhanced mass transport. The authors should compare the results obtained with the chronoamperometric pulses with other mass transport enhancement phenomena such as RDE or electrolysis under flow conditions.

Answer 7: According to the reviewer's kind suggestion, rotating disk electrode (RDE) experiments were carried out to understand the pulse function. Compared with the non-stir potentiostatic condition, the ammonia yield rate increases by ~60% at the stir potentiostatic condition (1800 rpm), indicating the existence of mass-transfer limitation for low-concentration nitrate electroreduction. Notably, the ammonia yield rate further increases under pulse conditions without stirring. These results prove that the pulse approach can not only weaken the mass-transfer limitation of low-concentration nitrate (**Fig. R17a, revised Supplementary Fig.27**) but also promote the thermodynamics of NRA. *In situ* ATR-FTIR experiments were performed to reveal the function of pulsed potential from the thermodynamic perspective. The extra bands at $\sim 1690\text{ cm}^{-1}$ and $\sim 1740\text{ cm}^{-1}$ (designated to on-top adsorption of NO (*J. Catal.* 2007, 249, 311-317; *Langmuir* 2008, 24, 4352-4357)) can only be observed under pulsed conditions (**Figure R17b, Fig.5d in the revised manuscript**). Based on the information offered by online DEMS, *in situ* ATR-FTIR, and reported literature, the possible reaction pathways were deduced as N-end pathway ($*\text{NO}_3 \rightarrow *\text{NO}_2 \rightarrow *\text{NO} \rightarrow *\text{NOH} \rightarrow *\text{N} \rightarrow *\text{HN} \rightarrow *\text{H}_2\text{N} \rightarrow *\text{NH}_3$) (**Figure R17c, Fig.5e in the revised manuscript**). NO has been considered as the critical intermediate in NRA. It has been reported that the reduction rate of on-top adsorbed NO is higher than that of bridged adsorbed NO (*J. Phys. Chem. C* 2010, 114, 6011-6018; *Langmuir* 2008, 24, 4352-4357; *Chem. Rev.* 2009, 109, 2209-2244.). Thus, pulsed electrolysis can promote the formation of a more active NO absorption configuration, enhancing NRA performance.

Figure R17. (a) The ammonia yield rate obtained under different conditions using a rotating disk electrode. The rotating rate was kept at 1800 rpm under stirred conditions. (b) The comparison of *in situ* ATR-FTIR spectra under the different conditions at $t = 300$ s. (c) Schematic illustration for the different NRA reaction pathways.

We acknowledge again for all the kind comments and wise suggestions from the third reviewer. We are sure that the quality of this work will be greatly improved according to these helpful comments and wise suggestions.

REVIEWERS' COMMENTS

Reviewer #1 (Remarks to the Author):

The manuscript has been revised according to the reviewer's requirements, and I agree to accept it in this version.

Reviewer #2 (Remarks to the Author):

The authors have revised the manuscript very well. This research could attract wide attention from scientists in the field of electrocatalysis. Thus, I suggest the publication of it in Nature Communications.

Reviewer #3 (Remarks to the Author):

The authors have carefully addressed all the referee comments and improved the quality of the manuscript significantly. The manuscript can now be accepted for publication.

A point-by-point response to the reviewers

To Reviewer 1:

Reviewer Letter:

The manuscript has been revised according to the reviewer's requirements, and I agree to accept it in this version.

Answer: We highly appreciate the reviewer for his/her positive comments on our revised manuscript. We are sure that the quality of this work has been greatly improved according to these nice comments and suggestions. Thanks very much.

To Reviewer 2:

Reviewer Letter: The authors have revised the manuscript very well. This research could attract wide attention from scientists in the field of electrocatalysis. Thus, I suggest the publication of it in Nature Communications.

Answer: We highly appreciate the reviewer for his/her positive comments on our revised manuscript. We are sure that the quality of this work has been greatly improved according to these nice comments and suggestions. Thanks very much.

To Reviewer 3:

Reviewer Letter: The authors have carefully addressed all the referee comments and improved the quality of the manuscript significantly. The manuscript can now be accepted for publication.

Answer: We highly appreciate the reviewer for his/her positive comments on our revised manuscript. We are sure that the quality of this work has been greatly improved according to these nice comments and suggestions. Thanks very much.